# Can Machine Learning Predict Stress Reduction Based on Wearable Sensors' Data Following Relaxation at Workplace? A Pilot Study

**Alessandro Tonacci [1],\*, Alessandro Dellabate [2],†, Andrea Dieni [2],†, Lorenzo Bachi [1], Francesco Sansone [1], Raffaele Conte [1] and Lucia Billeci [1]**

[1] Institute of Clinical Physiology-National Research Council of Italy (IFC-CNR), Via Moruzzi 1, 56124 Pisa, Italy; lorenzo.bachi@ifc.cnr.it (L.B.); francesco.sansone@ifc.cnr.it (F.S.); raffaele.conte@ifc.cnr.it (R.C.); lucia.billeci@ifc.cnr.it (L.B.)

[2] School of Engineering, University of Pisa, Largo Lucio Lazzarino 1, 56122 Pisa, Italy; alessandro.dellabate@live.it (A.D.); andrey2011@hotmail.it (A.D.)

\* Correspondence: atonacci@ifc.cnr.it; Tel.: +39-050-3152175

† These authors have contributed equally to the present work.

**Abstract:** Nowadays, psychological stress represents a burdensome condition affecting an increasing number of subjects, in turn putting into practice several strategies to cope with this issue, including the administration of relaxation protocols, often performed in non-structured environments, like workplaces, and constrained within short times. Here, we performed a quick relaxation protocol based on a short audio and video, and analyzed physiological signals related to the autonomic nervous system (ANS) activity, including electrocardiogram (ECG) and galvanic skin response (GSR). Based on the features extracted, machine learning was applied to discriminate between subjects benefitting from the protocol and those with negative or no effects. Twenty-four healthy volunteers were enrolled for the protocol, equally and randomly divided into Group A, performing an audio-video + video-only relaxation, and Group B, performing an audio-video + audio-only protocol. From the ANS point of view, Group A subjects displayed a significant difference in the heart rate variability-related parameter SDNN across the test phases, whereas both groups displayed a different GSR response, albeit at different levels, with Group A displaying greater differences across phases with respect to Group B. Overall, the majority of the volunteers enrolled self-reported an improvement of their well-being status, according to structured questionnaires. The use of neural networks helped in discriminating those with a positive effect of the relaxation protocol from those with a negative/neutral impact based on basal autonomic features with a 79.2% accuracy. The results obtained demonstrated a significant heterogeneity in autonomic effects of the relaxation, highlighting the importance of maintaining a structured, well-defined protocol to produce significant benefits at the ANS level. Machine learning approaches can be useful to predict the outcome of such protocols, therefore providing subjects less prone to positive responses with personalized advice that could improve the effect of such protocols on self-relaxation perception.

**Keywords:** autonomic nervous system; ECG; galvanic skin response; heart rate; heart rate variability; machine learning; mindfulness; neural networks; relaxation; signal processing; skin conductance; wearable sensors; yoga

## 1. Introduction

Psychological stress affecting mental and physical health is continuously increasing in nowadays' society, with several negative consequences on one's quality of life [1]. Individuals try to cope with

stress by following different strategies, from psychopharmacological to behavioral remedies, with alternating fortunes. However, the efforts to find non-pharmacological therapies to tailor stress and related disorders are continuously growing, and often take into account relaxation techniques, including yoga, mindfulness and other similar methods [2,3].

According to literature, yoga and mindfulness, for example, have proven beneficial effects on the autonomic nervous system (ANS) activity in several cohorts of patients. Indeed, despite a significant heterogeneity of the effects brought by the practice, mainly due to the different study populations, experimental settings and different techniques taken into account, beneficial effects were retrieved in several autonomic domains, including heart rate (HR) reduction, heart rate variability (HRV) increase and changes in blood pressure (see [3] for a review).

Specifically focusing on the ANS effects reported in young, non-diseased subjects, Sawane and Gupta [4] randomized a cohort of individuals into two groups, one of which performed yoga and the second one a swimming class. The authors found improvements in all autonomic parameters in both groups, with the subjects performing yoga asanas responding better on high frequency, suggesting increases in both ANS and parasympathetic activity. Other studies found just slight variations of ANS parameters, mostly limited to the HR [5,6], often selected for its ease of detection with respect to other parameters, including those related to the HRV.

In this framework, the use of wearable sensors was seen to be feasible, useful and minimally obtrusive for autonomic assessment in several studies [7,8], and in particular this approach was followed in stress monitoring and related perspective, with good success [9,10].

However, the continuous growth in relaxation techniques' spread enabled several employers to start adopting relaxation practices also at the workplace, often in very short sessions, undertaken in crowded, non-structured environments, with debatable efficacy and doubtful benefits to the health of the employees.

To this extent, to the best of our knowledge, no studies have investigated ANS activity using wearable sensors during a quick relaxation session performed in the workplace in young, non-diseased subjects.

In addition, relaxation protocols proposed to groups can be extremely efficient for some individuals, leaving other ones without appreciable effects in terms of well-being enhancement. To understand the psychophysiological specificities of each individual prior to the protocol administration could suggest specific personalized exercises to maximize the outcome of relaxation even in non-fully-structured environments. To this extent, the technological advancements in the domain of machine learning and artificial intelligence could represent useful aids to properly solve this issue in a quantitative, objective manner.

In light of all such considerations, in the aforementioned experimental setting, this pilot study aimed at discovering whether wearable, minimally invasive, solutions are able to detect changes related to the ANS activity during the presentation of a short video clip and of a short audio track related to the seven chakras of yoga in a cohort of young individuals without concomitant conditions. Furthermore, as an exploratory analysis, we also aim at investigating the potential usefulness of machine learning in discriminating, prior to the relaxation, the subjects which are more prone to receive positive effects from the protocol.

## 2. Materials and Methods

### 2.1. Study Population

For the present study, 24 healthy volunteers (5 males, 19 females, mean age 27.4 ± 5.5 years, age range 18–38) were enrolled. All subjects gave their informed consent for inclusion before they participated in the study. The study was conducted in accordance with the Declaration of Helsinki.

Exclusion criteria included the presence of associated cardiovascular or psychological/psychiatric conditions, usage of medicaments, or inability/unwillingness to sign informed consent.

## 2.2. Relaxation Procedure

The 24 subjects were randomly assigned to two groups, namely Group A and Group B.

After 3 min of resting, both groups were administered a relaxation protocol consisting of watching a video clip related to the 7 chakras of the yoga tradition on a PC screen (Task 1). The video clip, extracted from a 1-h long clip available on YouTube (San Bruno, CA, USA), lasted 210 s (3 and a half min) and included both audio and video tracks related to the 7 yoga chakras consecutively, with stimulation changing every 30 s (audio-video relaxation). The audio track was related to the classical sounds of each chakra, whereas the video was composed of 7 kaleidoscopes, each displaying the color of the corresponding chakra and containing the related symbol.

After this presentation, the subjects underwent a 3 min period of resting before undergoing the second track, lasting 3 and a half min, composed of the same video track as in Task 1 for Group A (video-only relaxation) and of the same audio track as in Task 1 for Group B but only a black screen in front of them (audio-only relaxation).

In summary, the overall protocol consisted of those five phases:
- Baseline (3 min): basal measurement. The subject, seated in a comfortable chair, was asked to relax during this phase;
- Task 1 (3 min, 30 s): presentation of audio and video relaxation protocol, as explained above;
- Inter-task (3 min): between-tasks resting state;
- Task 2 (3 min, 30 s): presentation of video (Group A) or audio (Group B) relaxation protocol, as explained above;
- Recovery (3 min): post-task basal measurement. The subject was asked to relax during this phase, similarly to the Baseline [7,11].

## 2.3. Signal Acquisition

Participants were equipped with devices for the acquisition of physiological signals, including electrocardiogram (ECG) and galvanic skin response (GSR). Both signals were acquired with unobtrusive wearable sensors manufactured by Shimmer Sensing, Inc. (Dublin, Ireland). More specifically, ECG was acquired through the single-lead, Bluetooth Shimmer ECG Unit at a sample frequency of 500 Hz, whereas GSR was captured at 51.2 Hz with the Shimmer3 GSR+ Unit according to a protocol already described elsewhere [12].

Both devices were connected by Bluetooth to a tablet, running a graphical user interface developed by Shimmer Sensing, Inc.

ECG and GSR signals were acquired during the five phases mentioned in the previous paragraph.

## 2.4. Psychological Questionnaires

The well-grounded, reliable [13] visual analogue scale for anxiety (VAS-A) [14] and state-trait anxiety inventory (STAI) [15] questionnaires were administered to the volunteers before and after the recording protocol to infer the state and trait anxiety for each of the subjects enrolled. Since STAI is composed of both STAI-Y1 and STAI-Y2, the first of which related to the state anxiety and the second one to the trait anxiety, only VAS-A and STAI-Y1 were repeated after the relaxation protocol.

## 2.5. Signal Analysis

### 2.5.1. ECG

The ECG signal was analyzed through a graphical user interface developed by our research group with Matlab (Mathworks, Natick, MA, USA), allowing extraction, from the raw signal, the associated tachogram according to the well-grounded Pan–Tompkins algorithm [16].

The interface also allows one to extract common time- and frequency-domain features associated with the signal [17,18], including:
- time-domain features:

- HR: number of heart beats occurring per time unit, expressed in bpm. The HR is normally related to the activity of the sympathetic branch of the ANS;
- Standard deviation of normal-to-normal intervals between two consecutive R peaks of the ECG signal (SDNN): measurement of the HRV, expressed in ms. SDNN is normally affected by both sympathetic and parasympathetic components of the ANS [19];
- Changes in successive normal sinus (NN) intervals exceeding 50 ms (pNN50), expressed as a percentage. Like other HRV measures, pNN50 also indicates the overall activity of the autonomic nervous system; however, under certain experimental conditions, pNN50 is often considered as a reliable indicator of the parasympathetic activity;
- Cardiac sympathetic index (CSI) extracted from the Lorenz plot. CSI is considered a reliable indicator for the sympathetic activity [20].

  - frequency-domain features:

- Normalized component of the ECG signal power spectral density at low frequency (0.04–0.15 Hz) (nLF). nLF is normally considered to be related to both sympathetic and parasympathetic activity;
- Normalized component of the power spectral density of the ECG spectrum at high frequency (0.15–0.4 Hz) (nHF). nHF is normally related to the parasympathetic activity;
- Low- vs. high-frequency components of the power spectral density of the ECG spectrum (Low-to-High Frequency (LF/HF) ratio). LF/HF ratio is often considered as a sort of balance between sympathetic and parasympathetic activity.

It is worth noting that all the frequency-domain parameters were extracted by the power spectral density estimated by the Welch method [21].

### 2.5.2. GSR

GSR signal was analyzed through the Matlab-based software Ledalab V3.4.9 (General Public License (GNU)) [22]. With the help of this tool, for each phase, the overall mean GSR signal and its tonic component were extracted. For this study, the phasic component was not considered since the study aimed at comparing the signal in the various experimental phases and not the single response to a given stimulation.

### 2.6. Statistical Analysis

In this study, statistical analysis was performed with SPSS v.23 (IBM Corporation, Armonk, NY, USA).

At first, we aimed to assess the normality of the variables' distribution using the Shapiro–Wilk Test [23].

In case of non-gaussianity, Friedman's test followed by Wilcoxon signed rank test was performed to compare the different phases, while Spearman's test for correlation analysis was applied. Correlations were further checked by applying the false discovery rate (FDR) test to control false positive cases.

### 2.7. Machine Learning

Based on the results from the statistical analysis, a machine learning approach was adopted using the dedicated Matlab App "Classification Learner". Several classifiers were trained using as input the autonomic features extracted from both ECG and GSR signal as described in Section 2.5.1 and Section 2.5.2. Such Matlab-based classifiers included tree, linear discriminant, quadratic discriminant, logistic regression, support vector machine (SVM) and k-nearest neighbor (KNN). The output was set to either "0" or "1" in case of "negative" or "no effect" or in case of "positive" effect of the relaxation protocol, respectively. Such value was based on the variation in the self-reported anxiety through STAI-Y1 scale: when the STAI-Y1 was increased after the treatment (i.e., increased anxiety), the output was set to "0", as in the case of no STAI-Y1 variation; conversely, when the STAI-Y1 score after the

protocol was decreased, the output was set to "1" (i.e., decreased anxiety). Concerning the methodology, we decided to use the cross-validation method as it appears to give a good estimate of the predictive accuracy of the final model trained with all the data. This approach requires multiple fits but appears to make efficient use of all the data, so it is recommended for small data sets. The best results were obtained using 5-fold cross-validation.

## 3. Results

### 3.1. Normality Test

According to the normality tests, all the variables were found to be distributed other than gaussian, therefore requiring all the statistical tests to be performed with non-parametric methods.

### 3.2. ECG Parameters

#### 3.2.1. Group A

As stated above, the 12 subjects of Group A underwent the protocol foreseeing audio+video and video only stimulation.

Here, both SDNN (F = 11.799, $p$ = 0.019) and CSI (F = 9.667, $p$ = 0.046) were significantly different between the test phases.

Specifically, SDNN was decreased at Task 1 with respect to the Baseline (Z = −2.040, $p$ = 0.041), with a following increase at Inter-task (Z = −2.118, $p$ = 0.034) and a subsequent decrease at Task 2 (Z = −2.001, $p$ = 0.045) (Figure 1a).

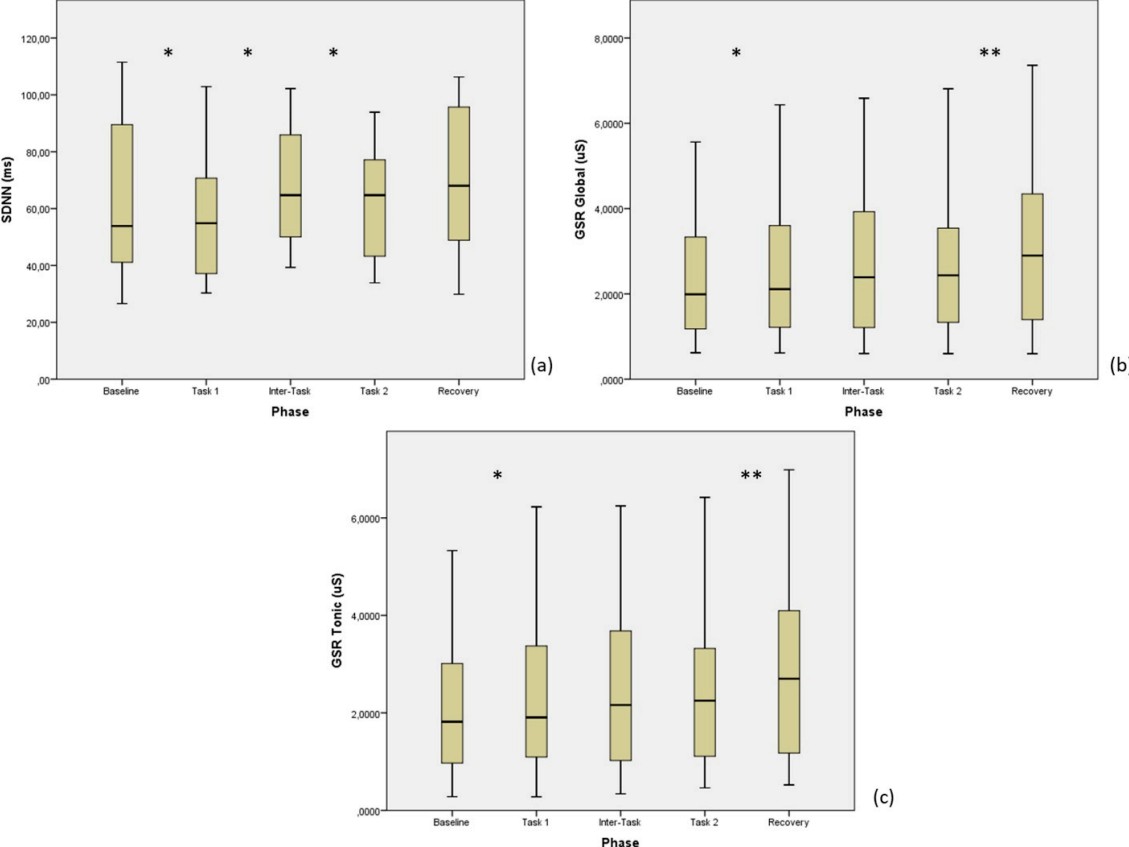

**Figure 1.** Autonomic parameters variation across the test phases for Group A: (**a**) SDNN; (**b**) global GSR; (**c**) tonic GSR (*: $p < 0.05$; **: $p < 0.01$).

As for CSI, no significant variations were seen comparing the single phases one-by-one.

### 3.2.2. Group B

No significant variations concerning ECG-related autonomic parameters were noticed for Group B, undergoing the audio+video and audio only protocol.

The results obtained, divided into the two groups, are reported in Table 1.

**Table 1.** ECG features, expressed as means ± SDs, for the two groups, separately.

| Group A | | | | | |
|---|---|---|---|---|---|
| Feature | Baseline | Task 1 | Inter-Task | Task 2 | Recovery |
| HR (bpm) | 72.6 ± 17.4 | 71.5 ± 15.6 | 71.8 ± 14.3 | 71.1 ± 13.6 | 72.6 ± 14.0 |
| SDNN (ms) | 64.3 ± 28.6 | 57.7 ± 24.5 | 67.4 ± 20.4 | 61.5 ± 20.0 | 70.7 ± 25.8 |
| pNN50 (%) | 25.3 ± 20.0 | 24.9 ± 22.5 | 25.6 ± 21.1 | 26.4 ± 22.7 | 25.7 ± 20.5 |
| CSI (ratio) | 2.41 ± 0.92 | 2.41 ± 1.03 | 2.64 ± 0.88 | 2.50 ± 1.02 | 2.68 ± 0.64 |
| nLF (n.u.) | 0.54 ± 0.23 | 0.52 ± 0.25 | 0.60 ± 0.23 | 0.54 ± 0.26 | 0.57 ± 0.18 |
| nHF (n.u.) | 0.46 ± 0.23 | 0.48 ± 0.25 | 0.40 ± 0.23 | 0.46 ± 0.26 | 0.43 ± 0.18 |
| LF/HF (ratio) | 1.90 ± 1.79 | 1.93 ± 1.96 | 2.71 ± 2.60 | 2.25 ± 2.51 | 1.93 ± 1.68 |
| Group B | | | | | |
| Feature | Baseline | Task 1 | Inter-Task | Task 2 | Recovery |
| HR (bpm) | 78.6 ± 11.0 | 77.1 ± 10.7 | 77.6 ± 8.8 | 77.1 ± 9.6 | 77.3 ± 8.8 |
| SDNN (ms) | 52.6 ± 25.5 | 51.7 ± 27.1 | 53.9 ± 29.7 | 53.6 ± 25.9 | 60.4 ± 32.4 |
| pNN50 (%) | 15.3 ± 13.6 | 15.7 ± 14.0 | 14.4 ± 12.9 | 15.0 ± 13.3 | 15.4 ± 14.6 |
| CSI (ratio) | 2.78 ± 0.65 | 2.63 ± 0.53 | 2.76 ± 0.65 | 2.81 ± 0.77 | 2.99 ± 0.92 |
| nLF (n.u.) | 0.56 ± 0.11 | 0.56 ± 0.12 | 0.55 ± 0.12 | 0.62 ± 0.13 | 0.58 ± 0.15 |
| nHF (n.u.) | 0.44 ± 0.11 | 0.44 ± 0.12 | 0.45 ± 0.12 | 0.38 ± 0.13 | 0.42 ± 0.15 |
| LF/HF (ratio) | 1.43 ± 0.80 | 1.41 ± 0.59 | 1.40 ± 0.78 | 1.93 ± 1.03 | 1.81 ± 1.49 |

### 3.3. GSR Parameters

### 3.3.1. Group A

The 12 subjects belonging to this group displayed variation in both global (Friedman's F = 14.533, $p$-value = 0.006) and tonic (F = 23.533, $p < 0.001$) GSR along the duration of the experiment.

Such variations consisted of an increased global and tonic GSR at Task 1 with respect to the Baseline (Wilcoxon's Z = 1.961, $p$ = 0.050 for global GSR; Z = 2.118, $p$ = 0.034 for tonic GSR), and a further increase for both parameters at Recovery with respect to Task 2 (Z = 2.981, $p$ = 0.003 for global GSR; Z = 3.059, $p$ = 0.002 for tonic GSR) (Figure 1b,c).

### 3.3.2. Group B

Subjects from Group B only displayed a slight significance concerning the differences in GSR signal. In particular, global GSR was changed along the experimental protocol (F = 9.533, $p$ = 0.049), with a particular increase at Recovery with respect to Task 2 (Z = 222.667, $p$ = 0.008) (Figure 2).

The results obtained, divided into the two groups, are reported in Table 2.

### 3.4. Questionnaires

Overall, Group A subjects displayed a reduction of anxiety on both questionnaires (Z = −2.226, $p$ = 0.026 for VAS-A, Z = −2.584, $p$ = 0.010 for STAI-Y1), whereas Group B subjects modified their performances on VAS-A only (Z = −2.271, $p$ = 0.023), without differences seen for STAI-Y1 (Z = −1.385, $p$ = 0.166).

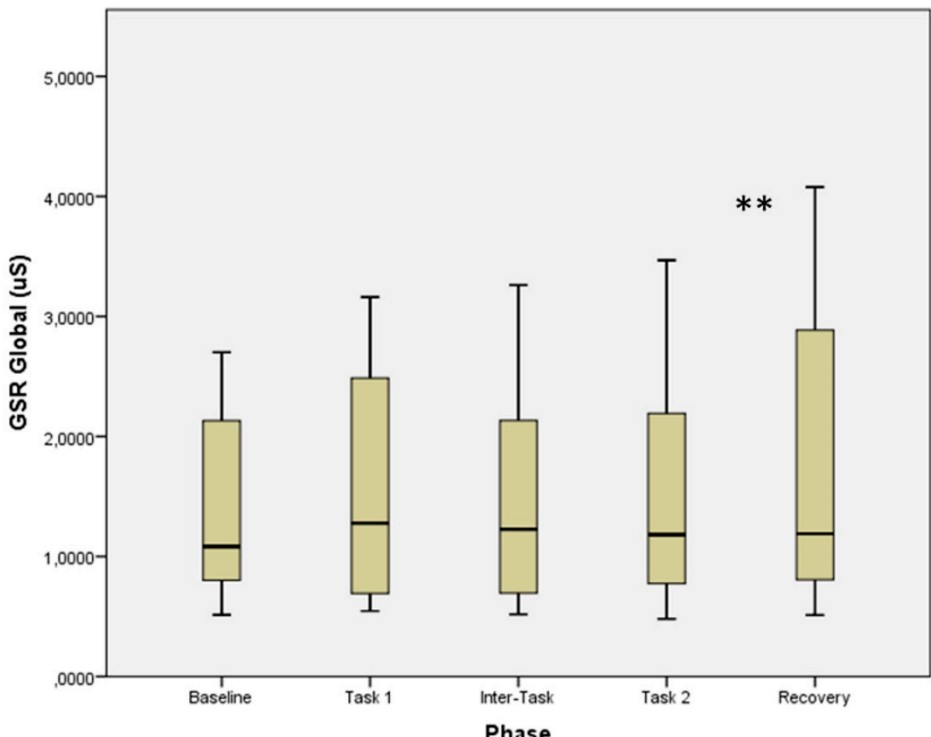

**Figure 2.** Autonomic parameters variation across the test phases for Group B: global GSR (**: $p < 0.01$).

**Table 2.** GSR features, expressed as means ± SDs, for the two groups, separately.

| Group A | | | | | |
|---|---|---|---|---|---|
| **Feature** | **Baseline** | **Task 1** | **Inter-Task** | **Task 2** | **Recovery** |
| **GSR Global (μS)** | 2.34 ± 1.54 | 2.54 ± 1.71 | 2.78 ± 1.88 | 2.80 ± 1.92 | 3.21 ± 2.11 |
| **GSR Tonic (μS)** | 2.10 ± 1.56 | 2.32 ± 1.73 | 2.54 ± 1.85 | 2.59 ± 1.87 | 2.98 ± 2.05 |
| **Group B** | | | | | |
| **Feature** | **Baseline** | **Task 1** | **Inter-Task** | **Task 2** | **Recovery** |
| **GSR Global (μS)** | 1.43 ± 0.82 | 1.57 ± 1.01 | 1.46 ± 0.93 | 1.52 ± 1.02 | 1.80 ± 1.31 |
| **GSR Tonic (μS)** | 1.31 ± 0.76 | 1.47 ± 0.97 | 1.37 ± 0.90 | 1.40 ± 0.99 | 1.65 ± 1.26 |

### 3.5. Correlations between Autonomic Parameters and Questionnaires

After applying the FDR, just a few correlations remained significant between autonomic parameters and questionnaires. In particular, significant positive correlations were seen between the difference between post-test vs. pre-test in GSR global and tonic and the difference between post-test vs. pre-test in VAS-A just in Group B subjects ($r = 0.641$, $p = 0.025$ for both GSR global and GSR tonic), whereas subjects belonging to the Group A did not display any correlations between autonomic parameters and questionnaires.

### 3.6. Machine Learning

Based on the two features more likely to discriminate between "positive" and "negative" or "no" effects of the relaxation protocol (i.e., CSI and LF/HF), several classifiers were trained taking advantage of the dedicated Matlab-based App.

Among them, the Subspace Discriminant classifier, trained on five cross validation cycles being the optimal trade-off between performances and computational load, was selected as the most effective one, providing a correct classification of the subjects in 79.2% of cases. The relative confusion matrix is displayed in Figure 3.

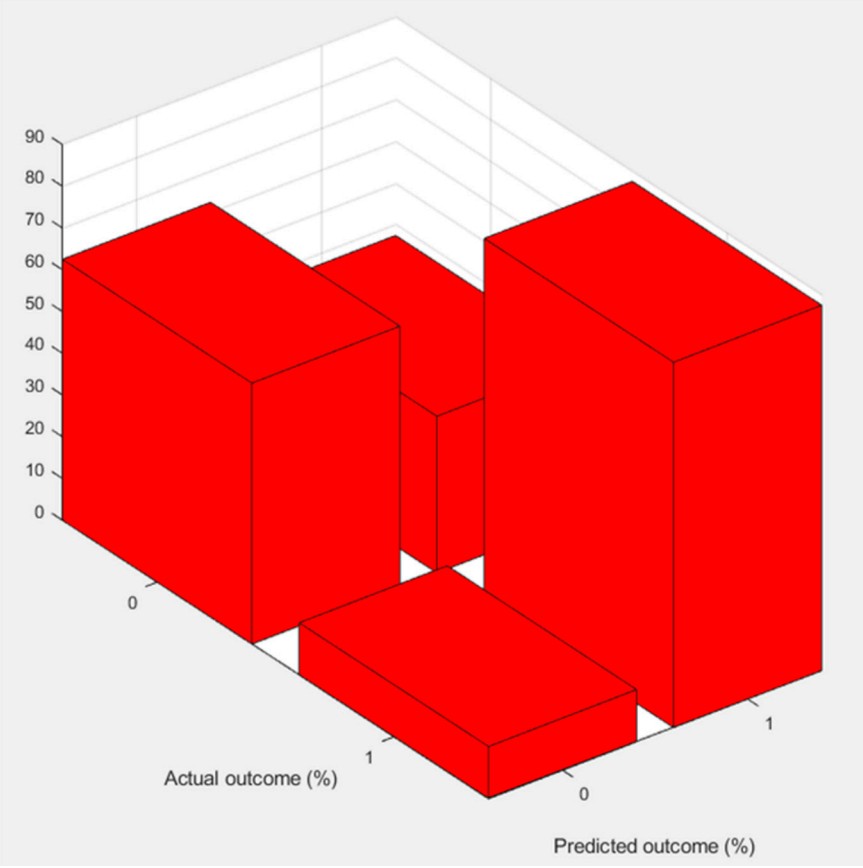

**Figure 3.** Confusion matrix for the selected classifier.

The data displayed revealed that the network correctly classifies the class "0" individuals (i.e., the subjects without a significant improvement of their well-being status) in 62.5% of cases, whereas the class "1" subjects (i.e., those self-reporting a significant well-being improvement) were correctly classified in 87.5% of cases.

## 4. Discussion

A quick, naturalistic relaxation protocol was seen to bring little difference in the autonomic domains studied here. Despite having reported a reduction in the perceived state anxiety of the majority of the subjects enrolled, confirmed by both VAS-A and STAI-Y1 results, the short duration of the protocol and, probably, the naturalistic experimental setting prevented the autonomic nervous system from displaying evident variations associated with the relaxation protocol. However, the trend noticed between the two groups was slightly different, driving to hypothesize a different stress level brought by the two distinct protocols to the volunteers.

More specifically, subjects from Group A displayed reduction in the HRV-related SDNN during the presentation of both audio+video and video-only stimulation, suggesting that such relaxing stimuli caused an overall autonomic reduction immediately during their presentation.

On the other side, the results obtained with the GSR analysis revealed an increased arousal occurring after the presentation of the stimuli, as highlighted by the higher GSR values at Recovery with respect to Task 2. A similar increase was also seen at Task 1 with respect to the Baseline, but only limited to Group A, possibly representing a false positive evidence of the protocol.

Interestingly, GSR measurement was correlated, in particular among Group B subjects, with the scores of the questionnaires related to state anxiety. In particular, subjects displaying higher increases (or lower decreases) of the perceived anxiety state also display higher increases of the GSR signal after the test completion. This fact demonstrated the coherence between some physiological signals related

to emotional stress and the perceived, self-reported stress scale related to anxiety state. This result confirms, and somewhat strengthens, previous evidence from existing literature highlighting the valuable contribution of GSR in detecting stress levels [24–26].

Concerning the variations of the autonomic parameters, despite heterogeneity between study protocols, existing literature is quite concordant about the positive effects on HRV brought by relaxation procedures. For example, positive HRV variations, in terms of HRV-LF decreases and HRV-HF increases, caused to healthy volunteers by mindfulness, were found by Nijjar and colleagues [27] and, earlier, by Takahashi et al. [28]. More recently, in more structured protocols, it was demonstrated that HRV can be considered as a reliable physiological marker for the capacity for self-regulation and adaptation [29], physiological characteristics were demonstrated to be enhanced by mindfulness protocols [30].

Similar benefits were also seen for yoga practice, especially those taking into account slow breathing protocols, as demonstrated by the review published by Nivethitha and colleagues [31].

Such autonomic benefits were seen for all subjects, independent of age [32,33], even though the magnitude of such effects appeared to be higher for older than for younger adults, according to Pal and colleagues [34].

In this regard, our protocol failed to replicate the majority of literature findings, probably because of the experimental setting adopted in the present study. Such results demonstrate that, despite a slight, yet significant variation in perceived stress scales brought by a brief relaxation protocol, in turn correlated with the GSR signal, more structured protocols should be administered by qualified trainers to allow detecting verisimilar autonomic changes referring to the beneficial effects of such relaxation practices.

As such, the use of wearables in characterizing the autonomic pattern of a person (or a group of persons) was seen to be highly acceptable by the end-user, providing useful information about the health and well-being status of the subject. Furthermore, they appear to be affordable—from a logistic point of view, featuring easy recharge, high portability and, somewhat, low cost—and reliable, with good stability of the signal acquired and easy data analysis and interpretation, as already seen in other works [7–10,35].

However, the key point of the present work, representing a further novelty of the approach described here, dealt with the use of machine learning tools for the prediction of the relaxation protocol outcome in this specific experimental setting.

To the best of our knowledge, this is the first scientific article to adopt this approach in the specific domain of audio and video relaxation protocols to evaluate their effect on perceived anxiety as main outcome. Indeed, just one recently published article, conducted on a large sample, went through this topic, limited to the administration of relaxation music, retrieving predictive factors that might influence therapeutic music listening outcomes [36].

In our research, the classifier adopted provided satisfying results, with classification accuracy near 80%. The vast majority (87.5%) of positive responders to the protocol were correctly classified, whereas among those not responding or negatively affected by the treatment, the correct classification fell to 62.5%. This fact is likely to be due to the low number of individuals belonging to the latter group, making the classifier training trickier and suggesting the need to enlarge the study population of this pilot to collect more data allowing this gap to be filled. This step is of paramount importance for our research since the main usefulness of the classifier is to identify a priori of possible non-responders to the treatment, customizing the protocol based on their specific needs.

In future, the proposed approach could be employed on a large scale to optimize the outcome and usefulness of relaxation treatments for stress reduction in workplaces.

*Limitations*

The results presented should be taken into account in light of some limitations. At first, a significant gender-bias is present when considering the overall study population. Indeed, a larger number of females are present with respect to males, accounting for a prevalence that could have masked

potential differential effects of the relaxation protocol based on gender. At the same time, the relatively small sample size, that is normal for a pilot study as is ours, has not been allowed to undergo further comparison (e.g., based on gender, age, etc.), discovering underlying differential effects of the relaxation at the autonomic level and to properly train the machine learning classifier, especially on non-responders. In such context, with small samples, the use of intra-person approaches, in which training and testing is performed on the same subject, could be a partial solution. However, we decided not to follow this option to stay even more conservative in our results.

At the same time, given the small cohort of subjects enrolled and tested, we only used one cross-validation value for assessing model performances. In future studies, on larger datasets, the application of diverse cross-validation values would reveal the best approach to be used based on the specific data included in the model.

Finally, another limitation concerns the methodology chosen for the relaxation that, in order to keep the protocol as simple and naturalistic as possible, was based on audio and video tracks downloaded from the popular web platform YouTube. Future studies, involving larger cohorts, should take into account the possibility of developing ad hoc protocols with the assistance of yoga or mindfulness qualified teachers.

## 5. Conclusions

The present study demonstrated the usefulness of the machine learning approach in identifying potential non-responders to relaxation treatments, which should be carefully considered when tailoring specific treatment protocols for stress reduction, based on their specific psychophysiological characteristics to optimize the treatment outcome. In addition, we demonstrated that wearables are able to detect autonomic changes eventually occurring during and after a relaxation protocol. However, this study also proved that, despite subjective beneficial effects already perceived by the subjects even after a short, non-structured relaxation procedure, a beneficial physiological response can be elicited only by more structured interventions that should be applied by experienced trainers in well-defined settings and locations.

This finding should be taken into consideration when undergoing protocols for stress reduction on both healthy and diseased subjects and can be possibly applied, with the proper methodology, in particular environments, including reduction of work-related stress and similar conditions.

**Author Contributions:** Conceptualization, A.T., R.C. and L.B. (Lucia Billeci); methodology, A.T and L.B. (Lucia Billeci); software, A.T., F.S., R.C. and L.B. (Lucia Billeci); validation, A.D. (Alessandro Dellabate) and A.D. (Andrea Dieni); formal analysis, A.T., A.D. (Alessandro Dellabate), A.D. (Andrea Dieni) and L.B. (Lucia Billeci); investigation, A.D. (Alessandro Dellabate) and A.D. (Andrea Dieni); resources, A.T. and L.B. (Lucia Billeci); data curation, A.T.; writing—original draft preparation, A.T., L.B. (Lorenzo Bachi) and L.B. (Lucia Billeci); writing—review and editing, A.T., L.B. (Lorenzo Bachi), F.S., R.C. and L.B. (Lucia Billeci); supervision, A.T.; project administration, A.T., R.C. and L.B. (Lucia Billeci). All authors have read and agreed to the published version of the manuscript.

**Funding:** This research received no external funding.

**Conflicts of Interest:** The authors declare no conflict of interest.

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
