# Peer review of "Can Machine Learning Predict Stress Reduction Based on Wearable Sensors’ Data Following Relaxation at Workplace? A Pilot Study"

_processes, doi:10.3390/pr8040448_

Round 1

Reviewer 1 Report

This paper presents an audio/video-based protocol aiming to evaluate stress reduction in the workplace. Particularly, machine learning algorithms are trained to classify the benefits of the proposed approach. It is a valid paper with a wide literature review and a clear description of the methods and results. However, there are some aspects to be taken into account in order to improve the quality of the manuscript:
1) Some acronyms are not defined (SDNN, RMSSD, pNN50). Authors should spell them out when appearing the first time.
2) Signal Acquisition: how did you choose the resting times (for baseline, Inter-task, and Recovery)? Please, clarify this aspect.
3) Psychological questionnaires: among all existing questionnaires, which criteria did you use to select VAS-A and STAY?
4) Machine Learning Algorithms: it is a reviewer’s suggestion to detail which particular classifiers have been trained.
5) Machine Learning Algorithms (k-fold cross-validation): Why did you choose k=5? Did you try other values? Which how would the results change by adopting other k values?
6) It is a reviewer’s suggestion to describe results in a separate section.
7) Results: The authors should define what F, p and Z values represent.
8) Authors should remove the unexpected white spaces in Pages 6 and 8 (e.g., by reducing the Figures size).

Author Response

We are grateful to the reviewer for their useful comments. Below their comments (in plain text), our point-by-point response (in italics), as well as eventual parts added to the text (underlined).

Reviewer 1

This paper presents an audio/video-based protocol aiming to evaluate stress reduction in the workplace. Particularly, machine learning algorithms are trained to classify the benefits of the proposed approach. It is a valid paper with a wide literature review and a clear description of the methods and results. However, there are some aspects to be taken into account in order to improve the quality of the manuscript:

Thank You for Your nice words and for Your useful comments.

1) Some acronyms are not defined (SDNN, RMSSD, pNN50). Authors should spell them out when appearing the first time.

Thank You. Now such acronyms are defined at their first mention.

2) Signal Acquisition: how did you choose the resting times (for baseline, Inter-task, and Recovery)? Please, clarify this aspect.

Thank You for Your observation. We followed the timing established in some published protocols dealing with ANS either by our research group and by other international groups (e.g., Tonacci et al., Sensors 2019; Van Puyvelde et al., Front Physiol. 2019). Those publications used a 3’ time for Baseline and Recovery phases, like we did in the present study. We also opted to not increase those sections beyond 3’ in order to not bring annoyance due to the eventual length of the assessment to the volunteers enrolled here. We included such two publications in our article, notably in Section 2.2.

3) Psychological questionnaires: among all existing questionnaires, which criteria did you use to select VAS-A and STAY?

Thank You. We decided to use them since, according to the literature, they are among the most reliable tests to assess anxiety, together with POMS and PANAS (see Rossi and Pourtois, Anxiety Stress Coping 2012 – reference added in Section 2.4), for their ease of administration and, especially in the case of STAI, for not suffering from the so-called ceiling effect, due to its composition upon two sub-scales of 20 items each that would demonstrate a somewhat variability in the distribution of results. Indeed, having used a scale with a too limited number of items would have hidden eventual small variabilities arising between the pre- and the post-test phase. Finally, with VAS and STAI tests, we were able to evaluate both state and trait anxiety, understanding the effect of the relaxation procedure on both those domains.

4) Machine Learning Algorithms: it is a reviewer’s suggestion to detail which particular classifiers have been trained.

Thank You. We added some more detail (see Section 2.7).

5) Machine Learning Algorithms (k-fold cross-validation): Why did you choose k=5? Did you try other values? Which how would the results change by adopting other k values?

Thank You for Your interesting observation. Indeed, we tried to use the most commonly employed k-value for cross validation, that is to say k=5 and k=10. However, possibly due to the low number of subjects involved in the present pilot, the results obtained with k=5 were better than those with k=10, suggesting us to apply this first option to our analysis.

In fact, according to the Matlab tool used for this purpose, with the cross-validation methodology, it is allowed to select the number of folds in which the dataset is partitioned (in the present study, k=5) and the data is partitioned into those 5 sets. For each set, the software trains the models desired using the out-of-fold observations, assesses model performance using in-fold data and calculates the average test error over all folds. Therefore, a higher value would have too much subdivided the data, given the fact that only N= 23 subjects were included in this pilot; conversely, a lower number of cross-validations would have impeded a proper data split.

6) It is a reviewer’s suggestion to describe results in a separate section.

Thank You. We added a very short section (3.1), citing the results concerning the normality tests.

7) Results: The authors should define what F, p and Z values represent.

Thank you. F is intended to be the Friedman’s Test statistics. Similarly, Z is the Wilcoxon’s Test statistics. P represents the p-value, meaning the probability of obtaining test results at least as extreme as the results actually observed during the test, assuming that the null hypothesis is correct. We specified the definition of those three indicators at their first mention (see “old” Section 3.2.1, now 3.3.1).

8) Authors should remove the unexpected white spaces in Pages 6 and 8 (e.g., by reducing the Figures size).

Thank You. At present, it seems looking fine. Please, advice us in case it is not, thank you.

Reviewer 2 Report

The paper is a pilot study on how to predict outcome of relaxation treatments by using machine learning methods. The experiments and the statistical analysis are well defined and the paper structure is clear. I have some minor concerns:

i understand this is a pilot but to obtain better results from ML tools it should be better to increase the number of participants, in order to have more data to train models (this should be included in the discussion)

do authors tried to use a intra-person approach (training and testing agorithms on the same subject - in diferent tests)? In a context like "stress reduction" (moreover when dealing with a low number of subjects) this may be a potential solution (maybe ad also this to the discussion) 

Author Response

We are grateful to the reviewer for their useful comments. Below their comments (in plain text), our point-by-point response (in italics), as well as eventual parts added to the text (underlined).

Reviewer 2

The paper is a pilot study on how to predict outcome of relaxation treatments by using machine learning methods. The experiments and the statistical analysis are well defined and the paper structure is clear. I have some minor concerns:

i understand this is a pilot but to obtain better results from ML tools it should be better to increase the number of participants, in order to have more data to train models (this should be included in the discussion)

Thank You for Your consideration, with whom we totally agree. Indeed, this statement was already added in the Limitations section (4.1), since we are aware about its paramount importance and the need for solving this point in future works.

do authors tried to use a intra-person approach (training and testing agorithms on the same subject - in diferent tests)? In a context like "stress reduction" (moreover when dealing with a low number of subjects) this may be a potential solution (maybe ad also this to the discussion)

Thank you. We also agree with this point. However, we decided not to follow this approach in order to stay more conservative about our results. We added this short explanation in the Limitations, too.

Reviewer 3 Report

This is a research article aimed to assess whether machine learning is able to predict stress reduction following relaxation at workplace. Features used for building the ML-based model are related to ECG and GSR and are measured by using wearable devices. My general comment is that the aim of the study, as well as the adopted methodology should be more clearly described. Specific comments are reported below.

ABSTRACT

- “and analyzed physiological signals related to the Autonomic Nervous System (ANS) activity, including Electrocardiography (ECG) and Galvanic Skin Response (GSR)”. I suggest replacing “Electrocardiography” with “Electrocardiogram” since the authors are referring to the signal.

INTRODUCTION

- Page 1. No new line after “life”.

- Page 1. “…to find non-pharmacologic”. Please replace “non-pharmacologic” with “non-pharmacological”

- Page 2. No new line after “several cohorts of patients”.

- Page 2. “The authors found improvements in all autonomic parameters in both groups, with the subjects undergoing yoga asanas responding better on high frequency, SDNN, RMSSD and pNN50 parameters, suggesting increases in both ANS and parasympathetic activity.” I suggest rephrasing the paragraph. At this level, SDNN, RMSSD and pNN50 have not been introduced yet and the reader may not be aware that they are related to Heart Rate Variability. Otherwise, add a paragraph introducing what HRV is.

- Page 2. “This pilot study aimed at discovering whether wearable, minimally invasive solutions are able to detect changes related to the ANS activity during the presentation of a short video clip and of a short audio track related to the seven chakras of yoga in a cohort of young individuals without concomitant conditions.” The aim here described is not clearly reflected in the title. Please rephrase. Moreover, features used for building the ML-based model are related to ECG and GSR and are measured by using wearable devices. Thus, I suggest providing a more focused introduction that includes also a paragraph concerning the use and the reliability of wearable devices in this context and in similar contexts.

MATERIALS AND METHODS

- Page 2. “…without any cardiovascular and/or psychological/psychiatric condition, nor under medicaments…” Please remove. This information is reported in the exclusion criteria.

- Page 3. “Participants were equipped with devices for the acquisition of physiological signals, including electrocardiogram (ECG) and galvanic skin response (GSR).” No new line after this sentence.

- Page 3. The five phases should be described in “Relaxation procedure” (Section 2.2).

- Page 3. Section 2.5.1. Heart rate (HR) has been already defined in the introduction. Do not define again the acronym.

- Page 4. Section 2.5.1. “R–R intervals” have not been introduced.

- Page 4. Section 2.5.2. “several characteristic features were extracted” is too generic. Please state which are the extracted features.

- Page 4. “Several classifiers were trained with the autonomic features as inputs.”. What does it mean “several classifiers”? Moreover, no definition of “autonomic features” has been given.

- Page 4. “To evaluate the performance of the classifiers, a 5-fold cross-validation was applied”. It is not clear which data has been used for training, validation and test. Cross-validation is a methodology usually applied for building the model, not to test performance.

RESULTS

- Page 8. Figure 3. It seems that the classifier provides a lot of false positive. Please comment.

DISCUSSION

- Page 10. “significant gender-bias is present when considering the study population, overall. Indeed, a larger number of females is present with respect to males”. Please introduce the information about gender in the description of study population.

- Page 10. "we demonstrated that wearables are able to detect autonomic changes eventually occurring during and after a relaxation protocol". Since this has been mentioned as a key point in the conclusion, the discussion should include a paragraph mentioning the reliability of wearable sensors in capturing important features related to autonomic activity, as already done in similar contexts:

Maranesi et al. Health monitoring in sport through wearable sensors: A novel approach based on heart-rate variability. Lecture Notes in Electrical Engineering 2016.

Author Response

We are grateful to the reviewer for their useful comments. Below their comments (in plain text), our point-by-point response (in italics), as well as eventual parts added to the text (underlined).

Reviewer 3

This is a research article aimed to assess whether machine learning is able to predict stress reduction following relaxation at workplace. Features used for building the ML-based model are related to ECG and GSR and are measured by using wearable devices. My general comment is that the aim of the study, as well as the adopted methodology should be more clearly described. Specific comments are reported below.

ABSTRACT

- “and analyzed physiological signals related to the Autonomic Nervous System (ANS) activity, including Electrocardiography (ECG) and Galvanic Skin Response (GSR)”. I suggest replacing “Electrocardiography” with “Electrocardiogram” since the authors are referring to the signal.

Thank You, our mistake. Corrected.

INTRODUCTION

- Page 1. No new line after “life”.

Thank You. Done.

- Page 1. “…to find non-pharmacologic”. Please replace “non-pharmacologic” with “non-pharmacological”

Thank You. Corrected.

- Page 2. No new line after “several cohorts of patients”.

Thank You. Done.

- Page 2. “The authors found improvements in all autonomic parameters in both groups, with the subjects undergoing yoga asanas responding better on high frequency, SDNN, RMSSD and pNN50 parameters, suggesting increases in both ANS and parasympathetic activity.” I suggest rephrasing the paragraph. At this level, SDNN, RMSSD and pNN50 have not been introduced yet and the reader may not be aware that they are related to Heart Rate Variability. Otherwise, add a paragraph introducing what HRV is.

Thank You, we agree with You. We decided to cut this part to make the sentence more concise and easier to the reader.

- Page 2. “This pilot study aimed at discovering whether wearable, minimally invasive solutions are able to detect changes related to the ANS activity during the presentation of a short video clip and of a short audio track related to the seven chakras of yoga in a cohort of young individuals without concomitant conditions.” The aim here described is not clearly reflected in the title. Please rephrase. Moreover, features used for building the ML-based model are related to ECG and GSR and are measured by using wearable devices. Thus, I suggest providing a more focused introduction that includes also a paragraph concerning the use and the reliability of wearable devices in this context and in similar contexts.

Thank You. We fully agree with Your observation. At first, we changed the title into “Can machine learning predict stress reduction based on wearable sensors’ data following relaxation at workplace? A pilot study” in order to highlight the importance of the use of wearables for our study. Then, we added a short consideration, with references, about the usefulness of wearables in evaluating autonomic parameters, overall, and stress in particular.

MATERIALS AND METHODS

- Page 2. “…without any cardiovascular and/or psychological/psychiatric condition, nor under medicaments…” Please remove. This information is reported in the exclusion criteria.

Thank You. We completely agree with You. We deleted this sentence.

- Page 3. “Participants were equipped with devices for the acquisition of physiological signals, including electrocardiogram (ECG) and galvanic skin response (GSR).” No new line after this sentence.

Thanks. Corrected.

- Page 3. The five phases should be described in “Relaxation procedure” (Section 2.2).

Thank You. Corrected.

- Page 3. Section 2.5.1. Heart rate (HR) has been already defined in the introduction. Do not define again the acronym.

Thank You. Corrected.

- Page 4. Section 2.5.1. “R–R intervals” have not been introduced.

Thank You. We better defined it in the same sentence.

- Page 4. Section 2.5.2. “several characteristic features were extracted” is too generic. Please state which are the extracted features.

Thank You, our mistake. We corrected the sentence accordingly.

- Page 4. “Several classifiers were trained with the autonomic features as inputs.”. What does it mean “several classifiers”? Moreover, no definition of “autonomic features” has been given.

Thank You. We integrated accordingly.

- Page 4. “To evaluate the performance of the classifiers, a 5-fold cross-validation was applied”. It is not clear which data has been used for training, validation and test. Cross-validation is a methodology usually applied for building the model, not to test performance.

Thank You for Your observation that let us better explain our approach. We decided to use the cross-validation method as it appears to give a good estimate of the predictive accuracy of the machine learning and it is recommended for small data sets. Thus, dealing with an exceptionally small dataset, we thought that applying cross validation on the entire dataset was acceptable, not dividing it into a training and test set.

We also added this part in Section 2.7.

RESULTS

- Page 8. Figure 3. It seems that the classifier provides a lot of false positive. Please comment.

Thank You for Your observation. Indeed, we discussed this point in the Discussion section (see end of Section 4, page 9-10): “In our research, the classifier adopted provided satisfying results, with a classification accuracy nearby 80%. The vast majority (87.5%) of positive responders to the protocol were correctly classified, whereas among those not responding or negatively affected by the treatment the correct classification fell down to 62.5%. This fact is likely to be due to the low number of individuals belonging to the latter group, making the classifier training trickier and suggesting the need to enlarge the study population of this pilot to collect more data allowing to fill in this gap. This step is of paramount importance for our research since the main usefulness of the classifier is to identify a priori possible non-responders to the treatment, customizing the protocol based on their specific needs.”

DISCUSSION

- Page 10. “significant gender-bias is present when considering the study population, overall. Indeed, a larger number of females is present with respect to males”. Please introduce the information about gender in the description of study population.

Thank You. We entered this information in Section 2.1, page 2, as: “24 healthy volunteers (5 males, 19 females, mean age 27.4 ± 5.5 years, age range 18-38), were enrolled for the present study.”

- Page 10. "we demonstrated that wearables are able to detect autonomic changes eventually occurring during and after a relaxation protocol". Since this has been mentioned as a key point in the conclusion, the discussion should include a paragraph mentioning the reliability of wearable sensors in capturing important features related to autonomic activity, as already done in similar contexts:

Maranesi et al. Health monitoring in sport through wearable sensors: A novel approach based on heart-rate variability. Lecture Notes in Electrical Engineering 2016.

Thank You. We agree with You, our mistake. We added a short consideration about this point in the Discussion section, with related references.

Round 2

Reviewer 1 Report

This paper presents an audio/video-based protocol aiming to evaluate stress reduction in the workplace. In detail, machine learning algorithms are trained to classify the benefits of the proposed approach. This new version of the manuscript has definitely been improved with respect to the previous one in terms of technical details and results description. Then, it is ready for being published.

Reviewer 3 Report

The authors have addressed all the issues raised in my previous review.
I do not have any further comment.